# Marine Ingredients for Sensitive Skin: Market Overview

**DOI:** 10.3390/md19080464

**Published:** 2021-08-17

**Authors:** Marta Salvador Ferreira, Diana I. S. P. Resende, José M. Sousa Lobo, Emília Sousa, Isabel F. Almeida

**Affiliations:** 1Associate Laboratory i4HB-Institute for Health and Bioeconomy, Faculty of Pharmacy, University of Porto, 4050-313 Porto, Portugal; msbferreira@ff.up.pt (M.S.F.); slobo@ff.up.pt (J.M.S.L.); 2UCIBIO–Applied Molecular Biosciences Unit, MedTech, Laboratory of Pharmaceutical Technology, Department of Drug Sciences, Faculty of Pharmacy, University of Porto, 4050-313 Porto, Portugal; 3CIIMAR–Centro Interdisciplinar de Investigação Marinha e Ambiental, Avenida General Norton de Matos, S/N, 4450-208 Matosinhos, Portugal; dresende@ff.up.pt (D.I.S.P.R.); esousa@ff.up.pt (E.S.); 4Laboratório de Química Orgânica e Farmacêutica, Departamento de Ciências Químicas, Faculdade de Farmácia, Universidade do Porto, 4050-313 Porto, Portugal

**Keywords:** marine ingredients, algae, sensitive skin, cosmetics

## Abstract

Marine ingredients are a source of new chemical entities with biological action, which is the reason why they have gained relevance in the cosmetic industry. The facial care category is the most relevant in this industry, and within it, the sensitive skin segment occupies a prominent position. This work analyzed the use of marine ingredients in 88 facial cosmetics for sensitive skin from multinational brands, as well as their composition and the scientific evidence that supports their efficacy. Marine ingredients were used in 27% of the cosmetic products for sensitive skin and included the species *Laminaria ochroleuca*, *Ascophyllum nodosum* (brown macroalgae), *Asparagopsis armata* (red macroalgae), and *Chlorella vulgaris* (microalgae). Carotenoids, polysaccharides, and lipids are the chemical classes highlighted in these preparations. Two ingredients, namely the *Ascophyllum nodosum* extract and *Asparagopsis armata* extracts, present clinical evidence supporting their use for sensitive skin. Overall, marine ingredients used in cosmetics for sensitive skin are proposed to reduce skin inflammation and improve the barrier function. Marine-derived preparations constitute promising active ingredients for sensitive skin cosmetic products. Their in-depth study, focusing on the extracted metabolites, randomized placebo-controlled studies including volunteers with sensitive skin, and the use of extraction methods that are more profitable may provide a great opportunity for the cosmetic industry.

## 1. Introduction

The largely unexplored marine environment harbors unique biodiversity and represents the vastest resource for the discovery of novel chemical entities with novel modes of action that cover a biologically relevant chemical space. These new scaffolds derived from various marine organisms offer valuable bioactive properties with great relevance in medical, pharmaceutical, and cosmetic fields [1,2,3,4,5,6]. Although synthetic strategies towards natural products have evolved tremendously over the last years, natural marine products are still preferred against their synthetic counterparts since they have better physicochemical, biochemical, and rheological characteristics, maintaining their stability at different pH and temperature ranges [7]. Among marine organisms, algae are recognized as one of the richest sources of new bioactive compounds [7]. The unique diversity of bioactive compounds contained in algae, such as vitamins, minerals, amino acids, sugars, lipids, and other biologically active compounds, is translated into numerous attractive properties for various industries [8], including the food, pharmaceutical, and cosmetic industries, as evidenced by the appearance in the market of various cosmetic products derived from these compounds [9]. Cosmetic products are stable substances or substance mixtures intended to clean, protect, perfume, and/or change the appearance of the external parts of the human body, teeth, and mucous membranes of the oral cavity, keeping them in good condition or correcting body odors [10]. They result from a formulation of raw materials which are categorized as active ingredients, excipients, and additives [11]. Cosmetic products may be categorized as body care, hair care, sun care, decorative cosmetics, oral care, and skin care, which is the largest cosmetic product category worldwide [12,13]. Skin care comprises a wide variety of products that should meet expectations of consumers with different skin types and organoleptic preferences. Sensitive skin is a condition characterized by multiple symptoms such as tightness, stinging, burning, or pruritus, which affects about 71% of the general adult population, being more frequent in the facial area [14,15,16]. Erythema, dryness, and desquamation are typically absent, but they may also occur [17]. Therefore, the sensitive skin segment allows meeting the needs of consumers who suffer from this condition [18]. Sensitive skin manifests in the presence of stimuli such as cold, heat, sun, pollution, cosmetics, or moisture which are not expected to produce unpleasant sensations, and the pathophysiological mechanisms involved in sensitive skin remain unknown [18]. Genetics, poor mental health, and microbiome imbalances have been proposed as contributing factors for this condition [19,20,21]. There are three hypotheses appointed in scientific literature for explaining the pathophysiology of this condition, namely the hyperactivity of the somatosensory and vascular systems, increased stratum corneum permeability, and an exacerbated immune response [22]. The hypothesis of an abnormal response from the somatosensory system is gaining increasing relevance. The skin contains sensory nerve fibers which are activated upon contact with physical and chemical stimuli such as heat, low pH solutions, or known irritants such as capsaicin, resulting in the release of neuropeptides, namely substance P or calcitonin gene-related peptide (CGRP). These neuropeptides cause a burning pain sensation through the activation of keratinocytes, mast cells, antigen-presenting cells, and T cells [23]. Neurosensory defects may lead to abnormalities in the communication with the central nervous system, resulting in a lower sensitivity threshold [15,16]. For example, an overexpression of transient receptor potential vanilloid type 1 (TRPV1), which is activated by heat and capsaicin, is thought to be involved in the pathophysiology of sensitive skin by increasing neuronal excitability [24,25]. Moreover, vascular hyperreactivity has been proposed in the pathophysiology of sensitive skin despite the absence of skin erythema [26]. The immune system is also associated with sensitive skin due to its interaction with nerve fibers, producing neurogenic inflammation. Neuropeptides activate keratinocytes, mast cells, antigen-presenting cells, and T cells, resulting in an inflammatory response [25]. Conversely, defects in the skin barrier function may be due to a derangement of intercellular lipids, due to a decrease in the ceramide content, a thinner stratum corneum with smaller corneocytes, as well as lower levels of pyrrolidone carboxylic acid (PCA) from the natural moisturizing factor (NMF), bleomycin hydrolase (BH), which is responsible for profilaggrin conversion, and transglutaminase (TG), which is essential for catalyzing the cross-linking between proteins and lipids during the corneocyte maturation process [22,27,28]. This may result in an increased permeability of the stratum corneum, which allows for the penetration of environmental aggressors [20]. More recently, this hypothesis has been questioned due to a study which failed to find significant differences in stratum corneum thickness, fatty acids, and the ceramide content, transepidermal water loss, or natural moisturizing factors between individuals with or without sensitive skin [29].

Recently, we have characterized the trends in the use of peptides in the sensitive skin care segment, reviewing their synthetic pathways and the scientific evidence that supports their efficacy [30]. We were able to conclude that three out of seven peptides have a neurotransmitter-inhibiting mechanism of action, while another three are signal peptides. As an example, palmitoyl tripeptide-8 may be a prime candidate for the development of pharmaceuticals aimed at alleviating the signs and symptoms of rosacea [30].

Given the variety of molecular targets involved in the sensitive skin pathophysiology, the chemical diversity of the marine ecosystem is a promising source for cosmetic ingredients for managing its symptoms. Our group has previously analyzed the market impact of marine ingredients in anti-aging cosmetics from multinational brands [31]. However, the use of these active ingredients in cosmetic products for sensitive skin remains unexplored. This study aims to unveil the state-of-the-art of marine ingredients in this segment by documenting their prevalence, as well as the most relevant species, their composition, and the scientific evidence that supports their efficacy for sensitive skin care. 

## 2. Trends in the Use of Marine Ingredients in Cosmetic Formulations for Sensitive Skin

The analysis of the presence of marine ingredients in all of the studied 88 cosmetic formulations for sensitive skin (19 multinational brands) indicated that 27% of them contain marine-derived ingredients. Interestingly, a more detailed analysis regarding the origin of these ingredients (Table 1) revealed that they all derive from algae, mainly macroalgae. Although over the last decades mariculture and aquaculture techniques have been developed towards the sustainable supply of other marine organisms, such as fish, sponges, corals, mollusks, echinoderms, *Artemia*, plankton, and microorganisms [5,11,32,33,34,35,36,37,38], their potential is not translated in the number of cosmetic formulations for sensitive skin that have been commercialized in the Portuguese market and contained the referred ingredients. The constraints related to reduced biomass availability of these marine organisms and difficulties regarding their production/cultivation at larger scales still represent a major bottleneck in the sustainable supply of the desired natural ingredients for the cosmetic industry [32].

On the other hand, algae are emerging as one of the most promising long-term, sustainable sources of bioactive ingredients to be used in the formulation of cosmetic and skin care products, with a large number and wide variety of benefits associated with their secondary metabolites [2]. Their biodiversity, easy cultivation, and growth modulation are the main reasons for their increased use in a variety of industries [2]. Depending on their size, they can be divided into macroalgae, which are seaweeds and other benthic marine algae that are generally visible to the naked eye, and microalgae, which require a microscope to be observed [39]. Additionally, macroalgae can be divided into three groups based on their dominant pigments: Rhodophyceae (red algae), Phaeophyceae (brown algae), and Chlorophyceae (green algae) [40]. Bioactive substances derived from these algae have diverse functional roles as a secondary metabolite and these properties can be applied to the development of novel cosmetic products. Brown algae account for approximately 59% of the total macroalgae cultivated in the world, followed by red algae at 40% and green algae at less than 1% [40]. Hence, it is interesting to notice that the wider availability of brown and red algae is clearly translated to their use as ingredients amongst the 88 studied cosmetic formulations for sensitive skin (Figure 1). Additionally, red algae are represented in 17% of the cosmetic formulations, and none of them contained green algae, probably due to their limited availability and, therefore, associated cost to the cosmetic industry. Microalgae are also well-represented, with 13% of the studied formulations containing these marine ingredients (Figure 1).

For several years, the use of the designation “algae extract” as a marine ingredient present in several cosmetic formulations with no specification of the species was permitted and included in the European Commission database for information on cosmetic ingredients contained in cosmetics (CosIng) [41]. Nowadays, the Commission requires that the new name assignment should be based on the current genus and species name of the specific alga. However, for an interim period of time, trade name assignments formerly published with the INCI name “algae extract” were retained. In this study, 2.3% of the 88 studied cosmetic formulations contained algae extract as a marine ingredient (Table 1, marked in Figure 1 as “undefined”); since the type of algae was not specified, a further detailed analysis could not be performed.

## 3. Efficacy of Algae-Containing Formulations on Sensitive Skin

The search results are summarized below (Figure 2):

### 3.1. Brown Macroalgae

Brown seaweeds belonging to two different taxonomic orders, Fucales (*Ascophyllum Nodosum*) and Laminariales (*Laminaria ochroleuca*), were used as ingredients in 17% of the studied cosmetic formulations. The brown color presented by these species results from the dominance of the pigment fucoxanthin (Figure 3), which masks the other pigments (chlorophyll *a* and *c*, β-carotene, and other carotenoids) and, as reserve substances, oils and polysaccharides [31]. The main polysaccharide found in the brown seaweeds is alginic acid, while laminarins (up to 32–35% dry weight) and fucoidans appear as sulfated polysaccharides (Figure 3).

Although these main constituents are common to both taxonomic orders Fucales and Laminariales, studies focused on the discovery of other secondary metabolites of *Ascophyllum nodosum* and *Laminaria ochroleuca* complemented with studies on the biological activity of these metabolites have also been developed and are analyzed below. The scientific and marketing evidence of the application of active ingredients from *Ascophyllum nodosum* and *Laminaria ochroleuca* in cosmetic formulations for sensitive skin was also compiled and analyzed.

*Laminaria ochroleuca* is a yellow brown digitate kelp presently distributed from Morocco to southwest England in the United Kingdom [42]. This species is highly sensitive to temperature, which models their growth and performance, and the recent ocean warming has led to a proliferation of *Laminaria ochroleuca* by extension of their geographical ranges to new habitats [43]. Due to these temperature and geographical changes, along with other variables such as habitat, season of harvesting, and environmental conditions (light, temperature, and salinity), this species experiences major shifts in their composition. An interesting study on the effect of different harvesting times, depths, and growth conditions of *Laminaria ochroleuca* revealed considerable differences in both qualitative and quantitative pigment profiles [44]. Significant seasonal variations in the photosynthetic pigment composition of *Laminaria ochroleuca* were observed which point to the occurrence of a photoprotective mechanism in the algae that deflects energetic resources to pigment biosynthesis. The samples collected in months with higher sun exposure (June–October) exhibited higher amounts of zeaxanthin, β-carotene, and chlorophyll *c* (Figure 4), with some species presenting nearly twice the levels of pigments, amongst which carotenoids were the most prevalent (56.1% of the total quantified) [44]. Another study dealing with the determination of phenolic compounds in *Laminaria ochroleuca* for human consumption revealed epigallocatechin (Figure 4) as the main polyphenol (760.2 ± 5.2 µg/g dry weight), followed by epicatechin (28.7 ± 2.0 µg/g dry weight), catechin gallate (21.4 ± 5.7 µg/g dry weight), epicatechin gallate (11.2 ± 1.6 µg/g dry weight), and epigallocatechin gallate (9.7 ± 1.3 µg/g dry weight) [45]. These polyphenols have been shown to provide an antioxidant, anti-inflammatory, and UVB protective action [46,47]. Other phenolic derivatives include linear phlorethols, containing either ortho, meta-, or para-oriented (or even a combination) C–O–C oxidative phenolic couplings (Figure 4) as exemplified in tetraphlorethols A and B [48].

A similar study was performed regarding fatty acid patterns of *Laminaria ochroleuca* [49]. This species exhibits a complex fatty acid profile, characterized mainly by the presence of medium and long fatty acyl chains (14–22 carbon atoms), with different degrees of unsaturation. The specimens from winter exhibited the lowest fatty acid concentrations (1255–1477 mg/kg of dry algae) whereas those harvested in warmer months presented higher fatty acid levels (1760 mg/kg of dry algae) [49].

Extracts of *Laminaria ochroleuca* have been incorporated in makeup, cleansers, moisturizers, and self-tanners, among other cosmetic products [50]. This extract is considered a natural skin soothing ingredient on several levels since it acts as an anti-inflammatory agent for skin irritations by boosting the skin’s immune response and protects the DNA from UV damage [51]. 

One raw materials supplier performed a transcriptomic analysis by mRNA extraction and evaluation of expression makers by RT-qPCR on reconstituted human epidermis (RHE model, 11 days old), after a single application of a lipidic *Laminaria ochroleuca* extract (3 mg/cm²) for 24 h [52]. An increase in the expression of proteins from the innate immune system was found, namely for toll-like receptor 4 (TLR 4), psoriasin (S 100 A7), RNAse 7, as well as upregulation of the enzymes linked to cellular homeostasis and oxidative stress, metallothioneins 1 (MT-1) and extracellular superoxide dismutase (SOD). Moreover, there was downregulation in the expression of proinflammatory cytokines IL-1α and IL-6, metalloproteinases 1, 3, and 9 (MMPs), as well as in plasminogen activator urokinase (PLAU), which are involved in the dermis’ extracellular matrix degradation.

The same supplier also performed a clinical study including 10 volunteers, who were exposed to a fixed irradiation dose of the minimal erythema dose (minimum dosage of radiation that produces skin erythema) × 1.5. Then, a gel formulation containing 2% *Laminaria ochroleuca* extract was applied to the test area, and another irradiated area was left untreated. The test area and the amount of product which was applied are not disclosed. Skin erythema was measured after product application and in the next 30, 60, and 120 min. The gel reduced skin erythema by 6.07% after 30 min, presenting the greatest difference in comparison to the control, and it kept reducing skin erythema over time. Statistical significance was not assessed.

Another lipidic *Laminaria ochroleuca* extract was evaluated by a distinct raw materials supplier regarding its biological activity in in vitro studies using reconstituted skin which was subject to epidermal trauma. After the extract application, an anti-inflammatory effect was observed through the inhibition of IL-1α and IL-6, as previously stated [52], but also through PGE_2_ release by epidermal cells and corneocyte degradation reduction, improving epidermal quality. Moreover, there was an increase in epidermal lipid content through phosphatidylcholine deposition, which contributes to reinforcing the epidermal barrier, thus reducing the penetration of environmental aggressors [53].

The anti-inflammatory activity of a lipidic *Laminaria ochroleuca* extract (Antileukine 6), which is mainly composed of phosphatidylcholine (Figure 4) derivatives, was evaluated in a murine model (C57BL/6 mice) [53,54]. Both ears were pretreated for 3 days twice a day with a *Laminaria ochroleuca* extract (2% in acetone/olive oil (4:1)) or the vehicle alone. Then, skin inflammation was induced by the application of 0.3% 2,4-dinitro-fluorobenzene (DNFB, hapten) in mouse ears, and the inflammatory response in terms of ear swelling (in μm) was scored at 0, 3, 6, 9, and 24 h in comparison with the vehicle applied at the other ear. The *Laminaria ochroleuca* extract reduced the inflammatory response as early as after 3 h, reaching the maximum effect at 6 h, with statistical significance, and showing a lasting effect up to 24 h. This anti-inflammatory effect may be due to the reduced DNFB penetration and/or a decrease in epidermal cytokines synthesis [54]. Having these results in mind, a *Laminaria ochroleuca* extract may be useful for reducing the symptoms associated with sensitive skin by improving the skin barrier function while modulating the neurogenic inflammation cascade by reducing the release of proinflammatory cytokines by mast cells, namely of IL-1 and prostaglandin E_2_ (PGE_2_). The metabolites which are responsible for these biological actions remain undisclosed.

*Ascophyllum nodosum* is an intertidal species characterized by its olive-brown fronds commonly detected around the periphery of the North Atlantic Ocean [55]. This intertidal fucoid has been extensively analyzed and studied for its chemical composition [55]. The most important constituents are the polysaccharides alginic acid, laminarins, and fucoidans (Figure 3), while other significant constituents like lipids, mannitol, ascophyllan, proteins, fibers, pigments, and phenols (Figure 5) [56,57,58], as well as vitamins, hormones, and enzymes are also present [55]. *Ascophyllum nodosum* has been used in bath oils, tablets, salts, as well as in skin cleansing and moisturizing cosmetics [50]. It has been shown to provide an antioxidant and photoprotective activity while inhibiting elastase and lipase [59,60,61,62]. While alginic acid or alginates have several applications in cosmetic formulations thanks to their thickening, gelling, emulsifying, and stabilizing abilities, fucoidans have been shown to reduce the intensity of the inflammatory response and promote a more rapid tissue healing, especially after wound or surgical trauma [55]. Fucoidans are fucans, sulfated polysaccharides with a fucose backbone, originating from seaweeds [39]. They have been shown to reduce the production of IgE by B cells which have been stimulated by allergens, thus blocking signals mediated by NFκB-p52. Furthermore, they have a free radical scavenging capacity, which may contribute to ameliorating skin inflammation [63,64]. Together, these properties make fucoidans a promising active ingredient for cosmetics intended to aid in the management of itching, stinging, and rashes [63]. One study evaluated the ability of this compound to reduce the inflammatory response using BALB/c mice as the murine model for atopic dermatitis and a DNFB solution (acetone/olive oil, 4:1) as the hapten [65]. Atopic dermatitis was induced in BALB/c mice by sensitization of the pre-shaved abdomen, with further challenge on the abdomen and ears after four, five, and nine days. Then, the treatment group received 50 μL of 0.2% fucoidan (from *Fucus vesiculosus*), while the negative control group received an acetone/olive oil vehicle, and the positive control group was given 0.1% dexamethasone. Fucoidan has been shown to ameliorate atopic dermatitis by decreasing inflammatory cell infiltration, splenocytes proliferation, and the CD4+ T cell response. However, fucoidans from *Ascophyllum nodosum* are distinct from those from *Fucus vesiculosus*, and their mechanisms of action are not expected to be effective on sensitive skin, based on what is known regarding the pathophysiology of this condition.

Ascophyllan has been shown to inhibit MMP expression, reduce the production of NO, tumor necrosis factor-α (TNF-α), and granulocyte colony-stimulating factor (G-CSF) more markedly than fucoidan and provide an antioxidant action [66,67]. No studies were found regarding its benefits for sensitive skin or inflammatory conditions.

One raw materials supplier evaluated the efficacy of a cosmetic formulation containing *Ascophyllum nodosum* and *Asparagopsis armata* (red algae) extracts [68]. Keratinocytes were exposed to phorbol myristate acetate (PMA), a tumor promoter and proinflammatory substance, and incubated with 0.2% of the active ingredient (methods are not further described) [69]. The incubated keratinocytes have shown a very significant reduction both in the vascular endothelial growth factor (VEGF) and PGE_2_ levels. VEGF stimulates the growth and dilation of capillaries, which may result in increased skin redness [17]. This combination also inhibited MMP-2 activity in a dose-dependent manner, reaching 37% inhibition at the concentration of 0.5%. Additionally, a clinical study was performed by the same supplier. Fifty-six volunteers presenting wrinkles and dry sensitive skin applied a formulation with 0.4% of this ingredient twice a day for 28 days. Then, their perception of the products was registered. Reduced tingling sensations, improved resiliency, immediate relief, and skin comfort were reported by 58%, 59%, 70%, and 71% of the volunteers, respectively. Although these results reveal a potential application of this ingredient for sensitive skin, it is not possible to conclude that *Ascophyllum nodosum* can be useful for this purpose as the tested ingredient also contains the algae *Asparagopsis armata*.

### 3.2. Red Macroalgae

*Asparagopsis armata* is a red seaweed which can be found in European coasts and in the Northeast Atlantic [70]. The main photosynthetic pigments of red algae are chlorophyll *a*, carotenoids (lutein, zeaxanthin, β-carotene) and phycobilins (phycocyanin and phycoerythrin), water-soluble pigments localized in the phycobilisomes, which give red algae their distinctive color [71]. Phycocyanin has been shown to provide anti-inflammatory, antioxidant, and wound-healing properties [72]. Besides photosynthetic pigments, red algae are also constituted of other interesting bioactive compounds (Figure 6), including agar [39], sulfated polysaccharides (carrageenans and porphyrans) [73,74], and mycosporine-like amino acids (MAA) [75,76,77,78].

The wide practical uses of these polysaccharides are based on their ability to form gels in aqueous solutions and act as a stabilizer, being generally used in creams, sticks, soaps, shampoos, lotions, foams, and gels [79]. On the other hand, MAAs are used in cosmetic formulations due to their photoprotective potential, antioxidant and skin protective properties [9,80,81]. They constitute a group of low-molecular-weight water-soluble molecules that can absorb UV radiation and disperse the absorbed energy as heat without generating reactive oxygen species (ROS), being a natural promising UV-absorbing alternative [82]. Examples of the most abundant MAAs in red macroalgae are mycosporine-glycine, shinorine, and porphyra-334 (Figure 6) [83]. The anti-inflammatory effects of these MAAs on the expression of genes associated with inflammation in response to UV irradiation was investigated using the human fibroblast cell line, HaCaT [82]. Mycosporine-glycine was able to suppress the expression of an inflammation marker gene, COX-2, in a concentration-dependent manner [82,83].

One raw materials supplier reported the cytostimulatory action of an *Asparagopsis armata* extract on human fibroblasts (WI 38), reaching the maximum level at 0.1%. No further details are provided [84]. 

Other studies including the use of an *Asparagopsis armata* extract in cosmetic formulations for sensitive skin were already disclosed [68,69]. However, these formulations contain not only *Asparagopsis armata*, but also *Ascophyllum nodosum*, and were previously described in Section 3.1.

### 3.3. Microalgae 

Marine microalgae also constitute an innovative source of bioactive compounds such as polyunsaturated fatty acids, tocopherols and sterols, vitamins and minerals, antioxidants, and pigments (e.g., chlorophyll and carotenoids), with great relevance in medical, pharmaceutical, and cosmetic fields [85]. Due to their unicellular or simple multicellular structure, they can grow rapidly and live under harsh conditions and environmental stressors such as heat, cold, anaerobiosis, salinity, photooxidation, osmotic pressure, and exposure to ultraviolet radiation [85]. The microalgae usually commercialized and used in biotechnology belong to the green algae, Chlorophyceae (such as *Chlorella vulgaris*, *Haematococcus pluvialis*, *Dunaliella salina*, and cyanobacteria) [85]. Their composition varies according to species and culture environments such as light intensity, temperature, pH, salinity, and medium [86]. *Chlorella vulgaris* is mainly constituted by proteins (43–58%), lipids (5–58%), carbohydrates (12–55%), pigments (chlorophyll (1–2%) and carotenoids (0.4%, astaxanthin, lutein, β-carotene, lycopene, canthaxanthin, see Figure 7 for examples)), vitamins (vitamins A, B, C, and E), and minerals (calcium, potassium, magnesium, and zinc) [86].

Sulfated polysaccharides from *Chlorella vulgaris* exhibited a capacity to prevent the accumulation and activity of free radicals and reactive chemical species, acting as protecting systems against these oxidative and radical stress agents [87]; in addition, peptides have been shown to reduce the matrix metalloproteinase-1 (MMP-1) expression in human skin cell fibroblasts, responsible for the breakdown of collagen [88]. The fact that a *Chlorella vulgaris* extract is able to stimulate collagen synthesis in the skin makes it suitable to be used in anti-aging cosmetics, as well as in wound-healing products [89,90]. 

Several studies report beneficial effects of extracts of *Chlorella vulgaris* for skin health. One study found that a *Chlorella vulgaris* extract was able to attenuate *Dermatophagoides farinae* (DFE)-induced atopic dermatitis (AD) in NC/Nga mice by oral administration [91], reducing 12-dimethylbenz[a]anthracene (DMBA)-induced tumor size and number by upregulating the sulfhydryl (-SH) and glutathione S-transferase (GST) levels in skin tissues [92]. These findings indicate *Chlorella vulgaris* could be useful as a preventive and therapeutic agent for various inflammatory skin diseases.

The evidence of the use of extracts of this microalga in cosmetic formulations for sensitive skin is limited. One cosmetic product was tested for its angiogenic inhibiting ability against positive and negative controls (suramide and VEGF, respectively) by using the in vitro model AngioKit^TM^ (TCS Cellworks), which allows following the development of the angiogenic process [93]. The formulation contained rhamnose, shea butter, argan oil, polyphenols, dextran sulfate, *Laminaria digitata*, caprapenols, *Chlorella vulgaris*, glycosaminoglycans, and UV filters (SPF 20). The formulation presented an antiangiogenic effect comparing to the positive control in the concentration range of 0.7–0.8 mg/mL, thus being useful for patients presenting rosacea. In spite of these results and *Chlorella vulgaris*’ potential to modulate the inflammatory response involved in sensitive skin, the composition of this formulation does not allow drawing conclusions regarding *Chlorella vulgaris*’ efficacy for treating this condition.

## 4. Materials and Methods

### 4.1. Data Collection

The composition of a pool of skin care facial cosmetic products from multinational manufacturers marketed in Portuguese parapharmacies and pharmacies was collected in 2019 in order to access the most used active ingredients in formulations for sensitive skin. Skin care products were included in the study if they exhibited in the label one of the following expressions: “sensitive skin” or “reactive skin” or “intolerant skin”. All the information available in the product labels was collected, along with the information available on the manufacturers’ websites. 

### 4.2. Data Analysis

The marine ingredients contained in cosmetic products for sensitive skin were listed according to the International Nomenclature of Cosmetic Ingredients (INCI). Afterwards, the data were analyzed with respect to the following parameters:

#### 4.2.1. Marine Ingredients Use 

The relative amount of cosmetic products for sensitive skin containing marine ingredients was evaluated and expressed in percentage.

#### 4.2.2. Top Marine Ingredients for Sensitive Skin

Marine ingredients were identified from INCI lists and ranked in the descending order of occurrence to disclose the top. Their categorization was also performed based on their marine organism species.

#### 4.2.3. Scientific Evidence Supporting the Efficacy of Marine Ingredients in Sensitive Skin Care

The efficacy data for each marine ingredient were searched in the online databases PubMed, Scopus, KOSMET, and SciFinder. Due to the lack of studies regarding the applicability of active ingredients in cosmetics for sensitive skin, a broader search was performed, using the keywords (“INCI name” OR “synonyms” when applicable) AND (“skin” OR “topical). 

## 5. Conclusions

Sensitive skin affects a significant proportion of the population worldwide, making it an appealing segment for the cosmetics industry. Marine organisms possess unique chemical pathways that are able to produce unprecedented scaffolds.

Marine ingredients were present in 27% of the analyzed cosmetic products for sensitive skin. Noteworthy, macroalgae are the prime marine ingredient used probably due to the easiness of cultivation allied with the development of a cutting-edge technology. These are easily cultivated either in a pond or a photobioreactor, in nonarable lands with minimal use of freshwater, or even in seawater or wastewater. It is also worth highlighting that among macroalgae, brown algae represent the main type of algae used in the analyzed cosmetic formulations.

Two preparations from brown algae (a *Laminaria ochroleuca* extract and an *Ascophyllum nodosum* extract), one—from red algae (an *Asparagopsis armata* extract), and one—from green microalgae (an *Chlorella vulgaris* extract) were found. The scientific evidence regarding the efficacy of these ingredients on sensitive skin is limited, especially due to the lack of clinical studies including volunteers with this condition. Noteworthily, there is one study that meets these requirements referring to a combination of an *Ascophyllum nodosum* extract and an *Asparagopsis armata* extract, which was found to reduce a tingling sensation, resiliency, and skin comfort in volunteers with sensitive skin. On the other hand, an *Laminaria ochroleuca* extract has a potential for improving the skin barrier function due to its lipid content and for reducing neurogenic inflammation by decreasing the release of pro-inflammatory cytokines by mast cells while increasing the production of antioxidant enzymes such as MT-1 and SOD. As for a *Chlorella vulgaris* extract, the in vivo evidence supporting its use in inflammatory conditions is still preliminary.

It is interesting to notice that efforts amongst the scientific community towards the identification of the active ingredient responsible for a certain property in the analyzed cosmetic formulation are still scarce. Usually, the entire extract is applied without further understanding of which chemical entity is associated with the bioactivity. Hence, research and development strategies should be employed both to identify the specific compounds responsible for the observed activities and determine their mechanisms of action. Among the chemical substances that can be found in these ingredients, carotenoids, sulfated polysaccharides, amino acids, and lipids are the most abundant. Of those, certain compounds usually isolated from marine organisms could be of interest for managing the symptoms of sensitive skin. Fucoidans from brown algae present evidence for managing inflammatory conditions, and they have been proposed to reduce itching and stinging symptoms. Additionally, mycosporine-like amino acids provide an antioxidant and anti-inflammatory activity by modulating the expression of the fibroblasts’ genes associated with inflammation. New strategies to increase the profitability of the extraction process are also needed in order to increase the cosmetic industry interest. Biotechnology may present advantages in this regard by reducing the environmental impact from the exploitation of these resources. The preliminary studies described herein are a major step towards the design of more innovative target-oriented ingredients by the cosmetic industry, providing efficacious products for sensitive skin. Overall, marine ingredients are already used in the sensitive skin segment, and they have a great potential to keep growing. Their in-depth study and the further investigation of other organisms, such as fish, sponges, corals, mollusks, echinoderms, *Artemia*, plankton, and microorganisms, constitute a great opportunity for formulators, cosmetic companies with R&D departments, and raw materials suppliers from the cosmetic industry.

## Figures and Tables

**Figure 1 marinedrugs-19-00464-f001:**
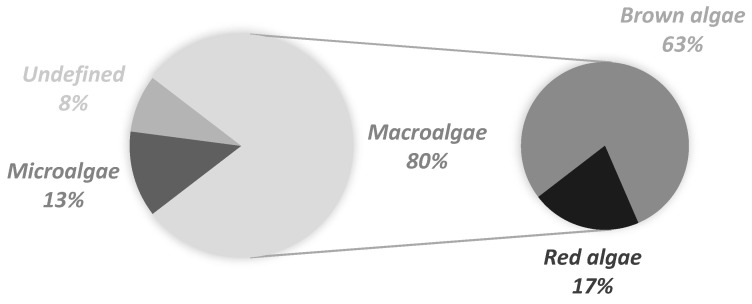
Categorization of the marine ingredients present in cosmetic formulations for sensitive skin commercialized in the Portuguese market (2019).

**Figure 2 marinedrugs-19-00464-f002:**
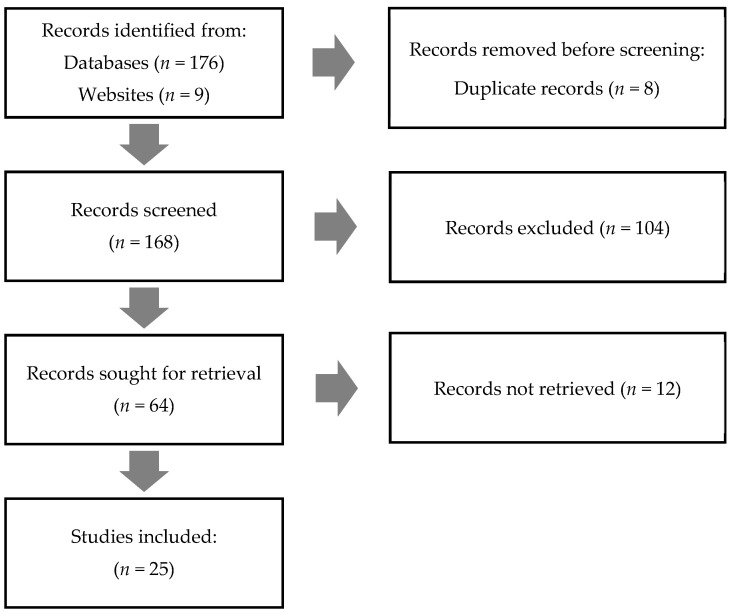
Flowchart of the selected articles according to four different parts of the search process: identification, screening, eligibility, and inclusion.

**Figure 3 marinedrugs-19-00464-f003:**
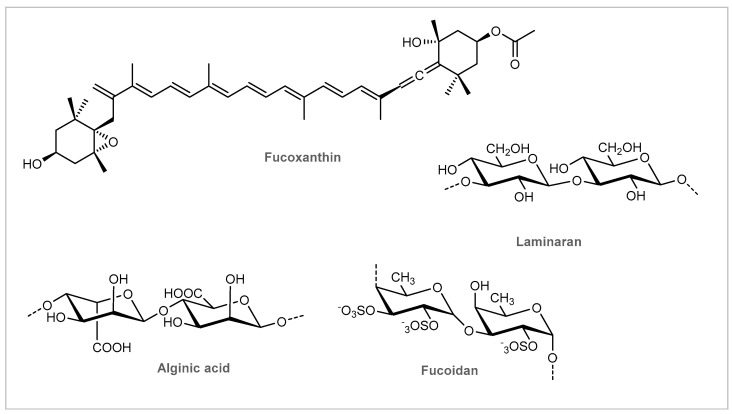
Bioactive constituents of brown seaweeds.

**Figure 4 marinedrugs-19-00464-f004:**
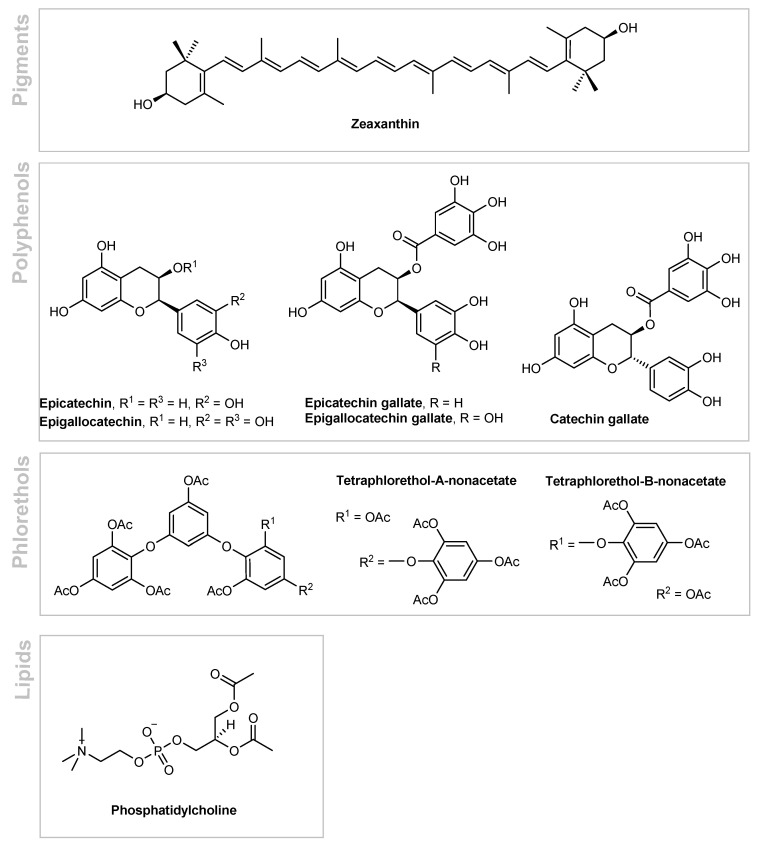
Bioactive metabolites of *Laminaria ochroleuca*.

**Figure 5 marinedrugs-19-00464-f005:**
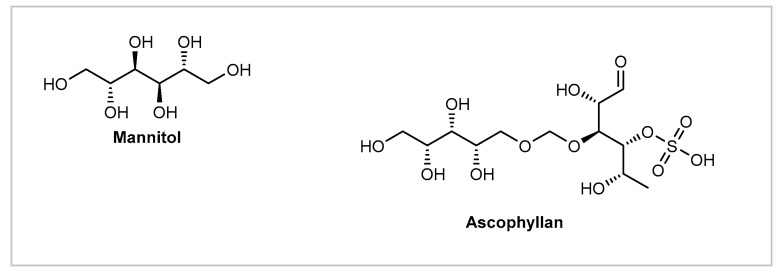
Bioactive metabolites of *Ascophyllum nodosum*.

**Figure 6 marinedrugs-19-00464-f006:**
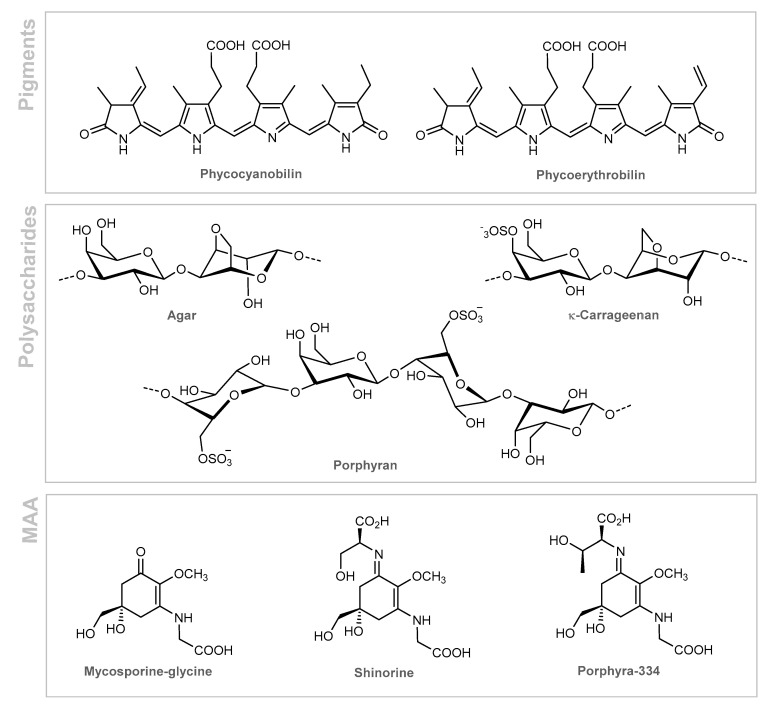
Bioactive constituents of red algae.

**Figure 7 marinedrugs-19-00464-f007:**
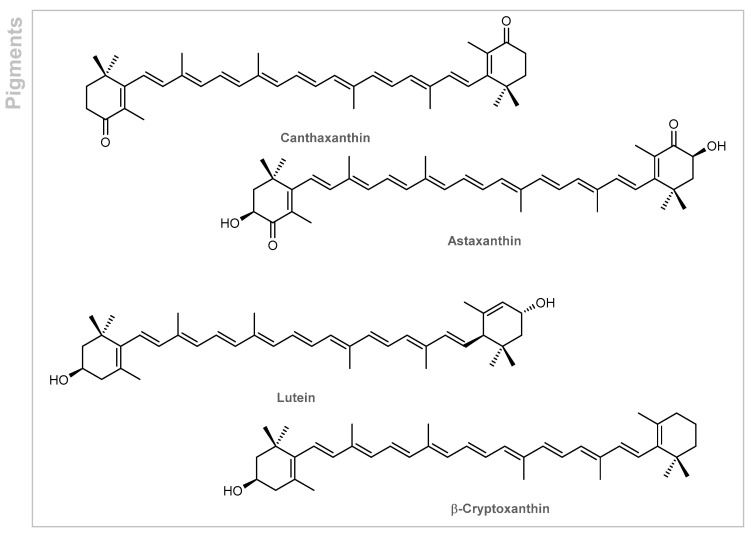
Pigments of *Chlorella vulgaris*.

**Table 1 marinedrugs-19-00464-t001:** Analysis of the prevalence and categorization of marine ingredients from the analyzed cosmetic products for sensitive skin (2019).

INCI ^1^	Category	*n*	%
*Laminaria ochroleuca* extract	Brown algae	11	12.5
*Ascophyllum nodosum* extract	Brown algae	4	4.5
*Asparagopsis armata* extract	Red algae	4	4.5
*Chlorella vulgaris* extract	Green microalgae	3	3.4
Algae extract	Undefined	2	2.3

^1^ INCI—International Nomenclature of Cosmetic Ingredients.

## Data Availability

Data sharing not applicable.

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
