# Peer review of "Marine Ingredients for Sensitive Skin: Market Overview"

_marinedrugs, 2021, doi:10.3390/md19080464_

Round 1

Reviewer 1 Report

The authors have performed a research on commercially available cosmetics for sensitive skin to identify products containing marine-derived extracts/compounds. They have classified the products according to their composition and provide data regarding potential clinical testing. The study aims to boost the incorporation and study of marine bioactive compounds in cosmetics.

I only have one very minor comment

On page 6 line 207 the authors state “genomic analysis by mRNA extraction” mRNA analysis is transcriptomic and not genomic

Author Response

Response to Reviewer 1 Comments

Point 1: The authors have performed a research on commercially available cosmetics for sensitive skin to identify products containing marine-derived extracts/compounds. They have classified the products according to their composition and provide data regarding potential clinical testing. The study aims to boost the incorporation and study of marine bioactive compounds in cosmetics.

I only have one very minor comment

On page 6 line 207 the authors state “genomic analysis by mRNA extraction” mRNA analysis is transcriptomic and not genomic

 Response 1: We acknowledge reviewer 1 for the kind comments. Sentence was corrected to “transcriptomic analysis by mRNA extraction”.

Author Response

Response to Reviewer 2 Comments

The paper entitled „Marine ingredients for sensitive skin: a market overview" (marine drugs-1346198) presents an overview of 88 products for facial skin care designed for sensitive skin, from Portuguese market.

Overall, I found this review very interesting. Authors did a good job and presented very nice and relevant literature regarding sensitive skin and the role of macroalgae (brown and red) and microalgae for sensitive skin care.

I only have a few suggestions for the authors:

Point 1: Figure 2 is not cited in text, please introduce it in the text.

Response 1: We acknowledge reviewer 2 for the correction. The sentence ”The search results are summarized below (Figure 2):” was added in line 159.

Point 2: It would be interesting to provide information concerning the categories of cosmetics that they have analyzed, their texture and if they had other claims besides the care of the sensitive skin.

Response 2: We acknowledge reviewer 2 for the kind suggestions. However, we believe that this analysis is outside of the scope of this work. We’ve analysed the ingredient lists of cosmetic products for facial care, regardless their texture, which had claims indicating they were formulated for sensitive skin, as we aimed to study the application of active ingredients from marine origin in these products.

Point 3: In introduction authors specify that the article analyzes the composition of cosmetics with ingredients for skin care from marine sources. A discussion about other ingredients present in these cosmetics would be interesting to see which type of symptoms characteristic to the sensitive skin may be relieved.

Response 3: We agree with reviewer 2 suggestion and the following paragraph was added to the introduction (line 94): “Recently, we have characterized the trends in the use of peptides in the sensitive skin care segment, reviewing their synthetic pathways and the scientific evidence that supports their efficacy [30]. We were able to conclude that three out of seven peptides have a neurotransmitter-inhibiting mechanism of action, while another three are signal peptides. As an example, palmitoyl tripeptide-8 may be a prime candidate for the development of pharmaceuticals aimed at alleviating the signs and symptoms of rosacea [30].”

We can only discuss about the usage of synthetic peptides in the sensitive skin care segment, since a previous study was already published by us. Discussion on other type of ingredients would take a deeper investigation, and we think that it is not the principal subject of this work.  As far as we know, no additional study regarding the use of other ingredients in cosmetic products for sensitive skin was published to date.

Point 4: Line 234 – Authors present the anti-inflammatory activity of a lipidic Laminaria ochroleuca extract in a murine model. Knowing that any cosmetic product or ingredient to be used as a cosmetic ingredient cannot be tested on animals, it would be better to use a reference which demonstrated this effect through an alternative method.

Response 4: We understand reviewer 2 concern. However, these are the only scientific evidences available in the literature to support the efficacy of these ingredients on sensitive skin. That doesn’t necessarily mean that the cosmetic industries responsible for the development of these products have tested either the ingredients or the final products on animals. Usually, these tests are performed using in vivo studies in human volunteers, since the used ingredients are already well-known.

Point 5: Line 270 – the same suggestion for the anti-inflammatory activity of fucoidans.

Response 5: Please see response 4.

Reviewer 3 Report

This paper analyses in detail market situation in the use of marine products for sensitive skin care. This study analyzed the use of marine ingredients in 88 facial cosmetics for sensitive skin from various  brands, and their composition and the scientific evidence that supports their efficacy.

There is an error in the figure 3 - the structure of fucoxanthin - missing quarternary carbon  between two double bonds (left part of molecule). Pls doublecheck the chirality of respective atoms - left part seems to be incorrect.

Fig. 3 pls. draw alginic acid in the chair type structure to conform other two structures (laminaran, fucoidan).

Figure 4 - incorrect structures of epigallocatechin and epigallocatechin gallate - R1   is incorrect - phenyl is missing

Phosphatidycholine - missing chiral hydrogen in the structure.

Figure 6 Pigments ... the structure left is not phycocyanin (which is a complex of the pigment-protein) - the structure presented there is actually phycocyanobilin - pls. correct.

Again Phycoerythin is a complex of protein with the pigment and the structure there is actually phycoerythrobilin

MAA - structure of shinorine is entirely incorrect - missing carbon (carboxyl) and missing chirality. Please, doublecheck all structures in this entire figure. There are still further errors.

The same for Fig. 7 doublecheck and add chiral atoms.

References need to be adjusted to conform journal style - please refer to recent paper in the journal and instructions. Please, make uniform writing capitals/small letters in paper titles - preferably make uniform small letters in whole references, please.

The whole paper must be carefully checked for typos and namely check all chemical structures. Remaining part of the paper seems to be quite OK.

Due to a large number of errors major revision would be appropriate and the paper will be reconsidered for potential acceptance.

Author Response

Response to Reviewer 3 Comments

This paper analyses in detail market situation in the use of marine products for sensitive skin care. This study analyzed the use of marine ingredients in 88 facial cosmetics for sensitive skin from various brands, and their composition and the scientific evidence that supports their efficacy.

Point 1: There is an error in the figure 3 - the structure of fucoxanthin - missing quarternary carbon between two double bonds (left part of molecule). Pls doublecheck the chirality of respective atoms - left part seems to be incorrect.

Response 1: We acknowledge reviewer 3 for the correction. Fucoxanthin structure in Figure 3 was corrected and the chirality was revised.

Point 2: Fig. 3 pls. draw alginic acid in the chair type structure to conform other two structures (laminaran, fucoidan).

Response 2: Alginic acid structure was changed in Figure 3 to be in the chair conformation.

Point 3: Figure 4 - incorrect structures of epigallocatechin and epigallocatechin gallate - R1   is incorrect - phenyl is missing

Response 3: We acknowledge reviewer 3 for the correction. However, after revision of the structures, we found that the structure of epigallocatechin is correct. Structures of epicatechin gallate and epigallocatechin gallate were incorrect (phenyl in R1 was missing) and were corrected in Figure 4.

Point 4: Phosphatidycholine - missing chiral hydrogen in the structure.

Response 4: Phosphatidycholine structure was corrected in Figure 4.

Point 5: Figure 6 Pigments ... the structure left is not phycocyanin (which is a complex of the pigment-protein) - the structure presented there is actually phycocyanobilin - pls. correct.

Response 5: The name of the structure was changed from phycocyanin to phycocyanobilin in Figure 6.

Point 6: Again Phycoerythin is a complex of protein with the pigment and the structure there is actually phycoerythrobilin

Response 6: The name of the structure was changed from phycoerythin to phycoerythrobilin in Figure 6.

Point 7: MAA - structure of shinorine is entirely incorrect - missing carbon (carboxyl) and missing chirality. Please, doublecheck all structures in this entire figure. There are still further errors.

Response 7: We acknowledge reviewer 2 for the correction. The structures of the MAAs in Figure 6 were entirely revised and corrected, with the addition of chirality.

Point 8: The same for Fig. 7 doublecheck and add chiral atoms.

Response 8: The structures contained in Figure 7 were entirely revised and corrected, with the addition of chirality.

Point 9: References need to be adjusted to conform journal style - please refer to recent paper in the journal and instructions. Please, make uniform writing capitals/small letters in paper titles - preferably make uniform small letters in whole references, please.

Response 9: References were adjusted to conform journal style (MDPI) and uniformized.

Point 10: The whole paper must be carefully checked for typos and namely check all chemical structures. Remaining part of the paper seems to be quite OK.

Response 10: The manuscript was entirely revised and all chemical structures were checked.

Point 11: Due to a large number of errors major revision would be appropriate and the paper will be reconsidered for potential acceptance.

Response 11: We acknowledge reviewer 3 for the important suggestions. We believe that they improved substantially the quality of the manuscript and we are thankful for that.

Round 2

Reviewer 3 Report

Authors amended all issues. The paper can be accepted for publication.